# The Influence of Subjective Socioeconomic Status on Life Satisfaction: The Chain Mediating Role of Social Equity and Social Trust

**DOI:** 10.3390/ijerph192315652

**Published:** 2022-11-25

**Authors:** Zirong Ren, Guoan Yue, Weilong Xiao, Qinghui Fan

**Affiliations:** 1College of Teacher Education, Zhejiang Normal University, Jinhua 321004, China; 2Key Laboratory of Intelligent Education Technology and Application of Zhejiang Province, Zhejiang Normal University, Jinhua 321004, China; 3Faculty of Information Technology, Monash University, Melbourne, VIC 3800, Australia

**Keywords:** subjective socioeconomic status, social equity, social trust, life satisfaction

## Abstract

Life satisfaction is significantly influenced by social capital, a key sociological term that links people to their social surroundings. Through a survey of 17,217 Chinese residents, this study investigated the probable processes of how subjective socioeconomic status affects life satisfaction within the framework of social capital. The results indicate that there is a positive correlation between subjective socioeconomic status and life satisfaction. Subjective socioeconomic status influences citizens’ life satisfaction not only through the independent mediating effects of perceived social equity and social trust, but also through the chain mediation of perceived social equity and social trust. This research advances our knowledge of the mechanisms behind the association between subjective socioeconomic status and life satisfaction. In improving citizens’ life satisfaction, we should not only provide sufficient subjective socioeconomic status to improve it, but also focus on the improvement of their social equity perceptions and social trust.

## 1. Introduction

Philosophers and psychologists have long been interested in the goodness of life and how to realize it [1,2,3]. The term “subjective socioeconomic status” refers to how someone is considered to be in terms of their place within society in relation to a particular reference group [4,5]. This is an important social measure that reflects a sense of self-within a social context and helps us understand social division and differentiation [6]. A growing number of research studies have found that subjective socioeconomic status shows more stable and stronger relations with psychological functioning (for example, negative effect, stress, and pessimism), health-related causes, awareness, empathy, and behavior than objective socioeconomic status [7,8,9,10]. When people feel higher subjective socioeconomic status, their alternatives of health, life satisfaction, happiness, and financial satisfaction strengthen, i.e., they experience higher life satisfaction [11]. As a component of social capital, less attention has been paid to how the different components of social capital work in between. This, this context will be examined in this study.

### 1.1. Subjective Socioeconomic Status and Life Satisfaction

Different socioeconomic groupings respond differently to the rise in material living standards brought on by economic expansion in terms of life satisfaction. Generally, the higher a person’s socioeconomic status, the higher the satisfaction with life. However, when an individual’s objective socioeconomic status, such as income and occupation, increases, their impact on life satisfaction decreases progressively. Subjective socioeconomic status, as an important adjunct to objective socioeconomic status, can be a greater meaningful positive predictor of life satisfaction [12]. Because subjective socioeconomic status shows how an individual compares to the group they are in, it is a better predictor of life satisfaction [13]. The theory of social comparison and the theory of reference groups suggest that reference groups play an important role in assessing an individual’s socioeconomic situation. Different reference groups may affect people in different ways [14,15]. Subjective socioeconomic status reflects the socioeconomic status felt by an individual relative to others in the social class [8]. According to relativistic theory, life satisfaction is more closely correlated with subjective socioeconomic status than with objective socioeconomic status [16]. The majority of studies have examined the correlation between absolute and relative income and life satisfaction to test this theory. Some studies have more closely correlated relative income with life satisfaction [17,18]. A meta-analysis study also confirms this, such that subjective socioeconomic status is more closely linked to life satisfaction [16]. Bunnk et al. found that comparisons with people of higher social status than themselves and those with lower social status than themselves may have a positive impact on individuals [19]. Based on the analysis, we propose Hypothesis 1: Subjective socioeconomic status can positively predict life satisfaction.

Depending on the function and the degree of subjective and objective perception, social capital can be divided into cognitive social capital, which includes social trust and social equity, and structural social capital, which includes social participation and social networks [20].

### 1.2. The Mediating Role of Social Trust

There are many ways to quantify social capital, yet trust in others is the one that is most frequently employed in Chinese study [21]. Trust is a functional social mechanism embedded in social structures and social institutions. High socioeconomic status is usually associated with high trust [22]. Lount and Pettit explored the impact of subjective socioeconomic status on trust. They argue that trusting parties’ perceptions of their high socioeconomic status leads to higher trusting behaviors than low socioeconomic perceptions [23]. Wilkinson and Goodman suggest that subjective socioeconomic status, as a category of socioeconomic status, provides better assessment information than the objective indicator of objective socioeconomic status and that it captures, more accurately, the more sensitive aspects of an individual’s socioeconomic status [24,25]. The positive predictive effect of subjective socioeconomic status on interpersonal trust has also been studied in cross-cultural studies [26]. Therefore, we believe that the subjective socioeconomic status of individuals may have a stronger relationship or predictive effect on social trust. The relationship between trust and life satisfaction has been proven in many studies [27,28,29]. This relationship is highly consistent, even in extreme situations, such as the worldwide serious money-based problem and natural disasters [30,31]. We assume that social trust acts as an intermediary between subjective socioeconomic status and life satisfaction.

### 1.3. The Mediating Role of Social Equity

The general equity theory holds that people act in ways that support, validate, and preserve the established order. They also tend to think that their own interests, the interests of their communities, and the interests of social institutions are good and moral. This reflects, among other things, the motivations for systemic, cluster, and self-legitimacy [32]. According to the status legitimacy theory, people from low-status groups are more likely than people from high-status groups to believe that society is fair [33]. A cross-country study including 36 nations found that people reported a fairer distribution of financial gain than individuals with a lower subjective socioeconomic status [34]. In an exceedingly transnational sample involving nineteen countries [35] and a sample of American residents with income stratification [36], subjective socioeconomic status also explicitly predicted perceptions of equity. It was also found that the perceived equity is surely predicted by the subjective socioeconomic status in the Chinese sample [37]. Additionally, a person’s level of live satisfaction was increased by their perception of equity [38]. Therefore, we infer that social equality mediates the link between subjective socioeconomic status and life satisfaction.

### 1.4. Social Equity and Social Trust

The “equality hypothesis” was put forth by politician Huntington, who contended that social equity would increase people’s degree of trust [39]. According to Rothstein and Uslaner, issues including inequality, corruption, unequal legal systems, and the informal sector decrease levels of social trust [40]. They argue that inequality needs to be reduced in order to raise social trust. Due to problems with unfair law enforcement and corruption, people would have a weaker perception of social equity, which will also affect their level of trust in the government [41]. Therefore, this study deduces that social equity and trust act as a chain intermediary between subjective socioeconomic status and life satisfaction. This study will construct a chain path of subjective socioeconomic status→social equity→social trust→life satisfaction.

### 1.5. The Present Study

In this study, we look at two key areas to see if the hypothesis is true. On the one hand, the current study aims to investigate the processes by which subjective socioeconomic status is related to life satisfaction. On the other hand, we wanted to find out if the relationship between subjective socioeconomic status and life satisfaction was mediated by social equity and social trust in parallel and in multiple ways. These ideas and the empirical data at hand led us to the following hypotheses:Subjective socioeconomic status is positively related to career satisfaction.Social equity and social trust are parallel mediators between subjective socioeconomic status and life satisfaction.Social equity and social trust are chain mediators between subjective socioeconomic status and life satisfaction.

All hypotheses are shown in Figure 1. Many empirical studies demonstrated that subjective socioeconomic status is associated with life satisfaction [13,42]. However, the internal mechanisms between subjective socioeconomic status and life satisfaction are still unknown. In addition to integrating multiple mediator variables and understanding the information between them, the multiple mediation model can comprehensively examine the complex processes and mechanisms of how independent variables affect dependent variables. Since the multiple mediation model has advantages over the simple model, this study constructs a multiple mediation model to explore the underlying mechanism between variables.

## 2. Materials and Methods

### 2.1. Samples and Data

This study uses data from CSS2017 and CSS2019, a large-scale continuous sampling survey project across China. The survey used a probability sample household visit method. The age range covered 17–70 years old (subject to the data at the time of the survey). The contents covered education level, personal work status, household products and living conditions, economic and social status, life satisfaction, interpersonal trust level, and trust level of government at all levels. The final sample size for model analysis was 17,217, and the SPSS20.0 software (IBM, Armonk, NY, USA) was used to calculate the relevant variables. A descriptive statistical analysis of the sample is shown in the Appendix A.

### 2.2. Variable Measurements

Subjective socioeconomic status. In this study, the subjective level of socioeconomic status of respondents was measured using the question “What level do you think your current socioeconomic status generally belongs to locally”. Using the Likert 5-point scoring method, 1 = “superior level”, 5 = “lower level”. We scored on this question in reverse, that is, the higher the score, the higher the subjective socioeconomic status.

Social trust. In this study, respondents’ general level of trust was measured by their agreement with the statement “I can trust most people in society” in a questionnaire. Using the 10-point scoring method, the higher the score, the higher the level of trust.

Social equity. The sense of social equity refers to an individual’s perception and evaluation of the distribution of social resources, opportunities, etc. We used eight questions to measure residents’ sense of social equity. Using the 5-point scoring method, the higher the score, the higher the sense of social equity. We used principal component analysis to extract a common factor and to generate a predictor score for “social equity”, which in turn generates a “social equity” variable.

Control variables. Research shows that demographics such as gender, age, and education level influence individual life satisfaction [43,44,45,46]. Therefore, age, education, sex, marital status, and place of residence were used as control variables in this study.

### 2.3. Common Method Deviation Control and Testing

In this study, there may be common methodological deviations in the data collected through the test self-presentation method. Using the Harman single factor test method, all entries were put together for exploratory factor analysis. When not rotated, the principal component factor analysis found that 5 factors could be extracted. The variance explanation percentage for the first factor was 22.56, which was lower than the 40% standard, indicating that there was no serious common method deviation in these data [47].

## 3. Result

### 3.1. Relevant Analysis

Descriptive statistics and correlation coefficients for all variables are shown in Table 1. According to the results of the national survey conducted in China, there is no gender-related difference in life satisfaction. Age, marital status, and place of residence all have a strong negative association. In other words, life satisfaction decreases with age and is also lower in divorced or widowed people than in other people. The household registration system, which makes agricultural household residents more susceptible to social injustice and inequality based on household registration, has a suppressive effect on people’s life satisfaction, and it may be the cause of rural residents’ lower life satisfaction than urban residents. Education level and life satisfaction are significantly and positively correlated, i.e., the higher the education level is, the higher the satisfaction with life; thus, it is important to improve the level of basic education for all people to improve the subjective life satisfaction of individuals. In China, residents’ life satisfaction is to some extent derived by comparing with others, and when seeing people around them having a poor life, relatively, their own life satisfaction will be stronger.

Subjective socioeconomic status is significantly positively correlated with social equity (r = 0.11, *p* < 0.001) and social trust (r = 0.17, *p* < 0.001), life satisfaction is significantly positively correlated with social equity (r = 0.21, *p* < 0.001) and social trust (r = 0.33, *p* < 0.001), and social trust is significantly positively correlated with social equity (r = 0.27, *p* < 0.001). All results are in line with theoretical expectations.

### 3.2. Mediation Effect Model

We used the PROCESS program in SPSS20.0 to analyze the chain intermediation path to test the promotion of equity of opportunity and social trust. Bootstrapping is set to 5000 resamples. If the 95% confidence interval (CI) of the mediation effect did not contain 0, the mediation effect was significant [48]. At the same time, we analyzed gender, age, education level, and marital status as excoriates. From Table 2 and Figure 2, the direct effect of subjective socioeconomic status on life satisfaction was 0.643, and the confidence interval was [0.610, 0.676]. In addition, subjective socioeconomic status had a significant indirect effect on life satisfaction, with an effect value of 0.136 and a confidence interval of [0.122, 0.150]. The effect value obtained by using social equity as the mediated variable was 0.037, and the confidence interval was [0.031, 0.044]. The effect value was 0.081 and the confidence interval was [0.07, 0.092]. The effect value obtained by using social equity and social trust as a chain intermediary was 0.018, and the confidence interval was [0.016, 0.022]. The effects of the three mediations were 4.750%, 10.398%, and 2.311%, respectively. The effect values of each model were all in the confidence interval, and the confidence interval did not contain 0.

Due to the significant correlation between age, education, marital status, place of residence, and satisfaction with life, these four variables were incorporated into the regression equation as covariables. The mediating effects were tested using 5000 repetitions of the bias-corrected nonparametric percentile. The results are shown in Table 3. The 95% confidence interval of each path did not contain 0, indicating the mediation effect of social equity and social trust and the chain mediation of social equity and social trust.

## 4. Discussion

Social capital is the sum of the actual or virtual resources that a person or a team obtains as a result of having an institutionalized and lasting network of mutual awareness and recognition [49]. Subjective socioeconomic status, social trust, and perceptions of social equity are all components of social capital. Based on CSS data in 2017 and 2019, this study used a multiple mediating factors model to experimentally evaluate the relationship between subjective socioeconomic position and life happiness and delved more into how social trust and equity function are a kind of mediator between them. Our findings suggest that elevating subjective socioeconomic status can enhance life satisfaction. According to previous studies, individuals with a low subjective socioeconomic status view life satisfaction as being lower than individuals with a higher subjective socioeconomic status [12,50]. Even after adjusting for the five research factors of age, sex, education, marital status, and location of residence, the results showed that subjective socioeconomic status significantly predicted life satisfaction. This finding suggests that subjective socioeconomic status was a significant external factor in enhancing life satisfaction. The subjective assessment of one’s own social and economic standing is known as subjective socioeconomic status [6]. According to the principle of social comparison, people can swiftly gather useful knowledge about themselves by comparing themselves to others [51]. Placing themselves in the group they are in and comparing themselves to others has a significant impact on how people evaluate their socioeconomic standing. When people believe that their socioeconomic standing is higher than that of their referents, they are more likely to experience psychological fulfillment [52,53]. The subjective socioeconomic status defined by social comparison has a greater impact on an individual’s life satisfaction than the objective socioeconomic status determined by objective circumstances, claims the theory of social comparison. This study’s findings also lend weight to that theory.

The study’s findings support social equity’s position as a mediator between subjective socioeconomic status and life satisfaction as well as the chain interaction between social equity and social trust in this relationship. According to the present research, social equity has a major impact on life satisfaction, meaning that people’s satisfaction increases when they are satisfied with all facets of societal performance and policies. Additionally, it is consistent with the idea of the just world belief (BJW), which holds that a fair world promotes people’s essentially random fate [54,55]. It has been demonstrated that BJWs increase life satisfaction [55].

BJW can boost people’s propensity to interact, trust, and believe in social order and others. It can also encourage consultation and problem solving [56,57,58]. Trust can be utilized to link subjective socioeconomic position to life satisfaction not just as an outcome variable but also as an intermediary mechanism. Social trust can significantly enhance people’s quality of life when it increases. According to our study’s findings, social trust has a significant predictive impact on life happiness, which is in line with the findings of other earlier studies on the topic [59,60,61]. This demonstrates that the influence of subjective socioeconomic status on life satisfaction has an intermediary path of socioeconomic status, social trust, and life satisfaction.

The present research further demonstrates that the link between subjective socioeconomic status and life satisfaction is mediated by a network of social equity and social trust. The more people feel that their subjective socioeconomic level is high, the more they will build social trust and thus increase their life satisfaction with a sense of social justice. The reserve capacity concept states that those with poor subjective socioeconomic status have fewer psychosocial resources, which has negative effects on their physical and mental health [12,62]. In situations where subjective socioeconomic status is determined through social comparison, people with high subjective socioeconomic status typically view society as fair, while the opposite is true for those with low subjective socioeconomic status. Social equity thus influences social trust in a positive or negative way, leading to either high or low life satisfaction as a result of social trust. This creates a chain reaction between subjective socioeconomic status, social equity, and social trust.

Theoretical and applied ramifications of this study are present. This study first extends and validates the reserve capacity model, the just world concept, the systemic equity theory, and the relativistic hypothesis. Second, it appears that life satisfaction and subjective socioeconomic status have a favorable association. Social trust and equity both function as ongoing and independent mediators. It is possible to make people feel more satisfied with their lives by improving their assessments of their subjective socioeconomic status and by implementing specific tactics to boost their feelings of social justice and trust.

This study is cross-sectional in nature. It is impossible to definitively prove that two variables are causally related. Experimental research or longitudinal data will need to be further validated in the future. Additionally, because every variable in this study was assessed subjectively, it is impossible to eliminate errors brought on by social acceptance effects. Future research may consider gathering information from a variety of sources to increase measurement accuracy.

## 5. Conclusions

This research looked at the link between subjective socioeconomic position and life satisfaction as well as the chain mediation functions of social trust and equality between Chinese adults. The findings demonstrate that subjective socioeconomic position was associated with life satisfaction in a favorable way. Additionally, social equity and social trust played a dual role in regulating the relationship between subjective socioeconomic position and life satisfaction—one acting independently and the other acting as a link in the chain.

## Figures and Tables

**Figure 1 ijerph-19-15652-f001:**
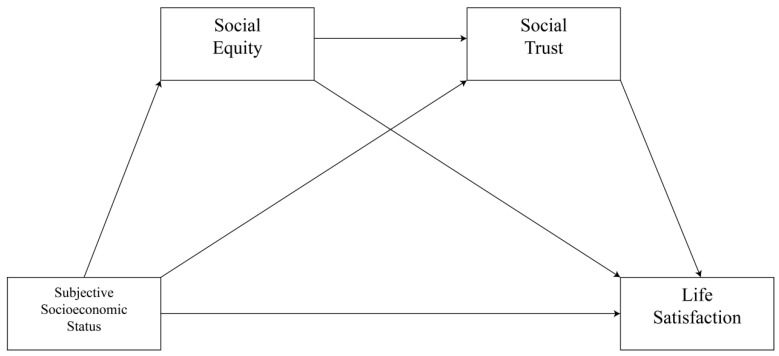
The multiple mediation model.

**Figure 2 ijerph-19-15652-f002:**
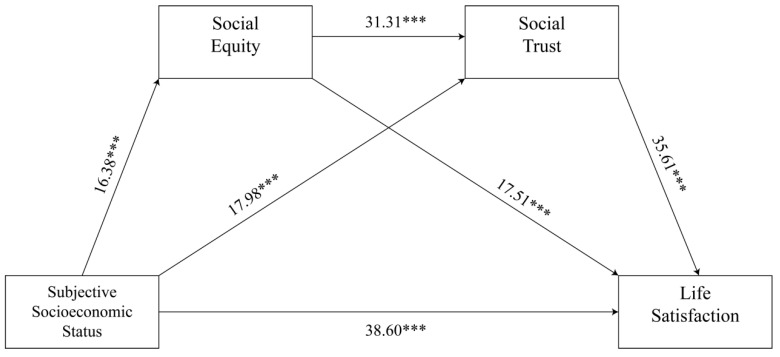
The pathway coefficients of multiple mediation models. *** *p* < 0.001.

**Table 1 ijerph-19-15652-t001:** Descriptive statistics and related analysis results for each variable.

	M ± SD	1	2	3	4	5	6	7	8	9
1. Gender	1.55 ± 0.50									
2. Age	45.78 ± 14.31	−0.04 ***								
3. Edu	3.73 ± 2.12	−0.08 ***	−0.50 ***							
4. Merial Status	2.09 ± 0.82	0.07 ***	0.39 ***	−0.27 ***						
5. Residence	4.13 ± 2.61	−0.01	0.14 ***	−0.36 ***	0.05 ***					
6. Social Equity	2.88 ± 0.61	0.04 ***	0.15 ***	−0.14 ***	0.04 ***	0.11 ***				
7. Social Trust	6.24 ± 2.04	−0.01	0.04 ***	−0.002	−0.01	0.07 ***	0.27 ***			
8. Life Satisfaction	6.99 ± 2.20	−0.01	−0.03 ***	0.13 ***	−0.07 ***	−0.04 ***	0.21 ***	0.33 ***		
9. SSS	2.23 ± 0.92	0.04 ***	−0.02 *	0.15 ***	−0.05 ***	−0.03 **	0.11 ***	0.17 ***	0.34 ***	

Note: * *p* < 0.05, ** *p* < 0.01, *** *p* < 0.001.

**Table 2 ijerph-19-15652-t002:** Analysis of the mediating effects of social equity and social trust between subjective socioeconomic status and life satisfaction.

Effect	Path	Estimate	*SE*	z	95% CI	Amount of Effect %
Direct effect	X → Y	0.643	0.017	36.647 ***	[0.610, 0.676]	82.542
Intermediary effect	X → M1 → Y	0.037	0.003	10.665 ***	[0.031, 0.044]	4.750
	X → M2 → Y	0.081	0.006	14.605 ***	[0.070, 0.092]	10.398
	X → M1 → M2 → Y	0.018	0.002	12.171 ***	[0.016, 0.022]	2.311
Total intermediation effect		0.136	0.007	19.111 ***	[0.122, 0.150]	17.458
Total effect		0.779	0.017	42.966 ***	[0.745, 0.813]	100

*SE* = *β*_sex_ + *β*_edu_ + *β*_marie_ + *β*_age_ + *β*_res_ + a1 * sss. Trust = *β*_sex_ + *β*_edu_ + *β*_marie_ + *β*_age_ + *β*_res_ + a2 * sss + d12 * *SE*. Life Satisfaction = *β*_sex_ + *β*_edu_ + *β*_marie_ + *β*_age_ + *β*_res_ + c. * sss + b1 * *SE* + b2 * trust. Indirect_All = a1 * b1 + a2 * b2 +a1 * d12 * b2, Ind_X_M1_Y = a1 * b1, Ind_X_M2 = a2 * b2, Ind_X_M1_M2_Y = a1 * d12 * b2, Direct = c., Total = c + a1 * b1 + a2 * b2 + a1 * d12 * b2, *** *p* < 0.001.

**Table 3 ijerph-19-15652-t003:** Regression analysis of the mediation model of social equity and social trust between subjective socioeconomic status and life satisfaction.

Var	Social Equity			Social Trust			Life Satisfaction		
	*β*	*SE*	*t*	*β*	*SE*	*t*	*β*	*SE*	*t*
Con	6.20	0.10	61.61 ***	4.01	0.12	32.77 ***	3.03	0.12	25.33 ***
Age	0.11	0.001	7.86 ***	0.01	0.001	5.22 ***	0.01	0.001	4.00 ***
Edu	0.16	0.01	16.20 ***	0.17	0.01	18.30 ***	0.15	0.01	16.70 ***
Merial Status	−0.15	0.02	−6.96 ***	−0.14	0.02	−6.26 ***	−0.12	0.02	−5.64 ***
Res	0.004	0.01	0.55	−0.01	0.01	−1.54	−0.02	0.01	−3.81 ***
Social Equity				0.81	0.03	30.13 ***	0.54	0.03	20.12 ***
Social Trust							0.31	0.01	39.56 ***
R^2^	0.05			0.09			0.22		
Adj.R^2^	0.05			0.09			0.22		
*F*	93.49			260.27			497.74		

Note: Var, variable; Edu, education; Res, residence. *** *p* < 0.001.

## Data Availability

The data used to support the findings of this study can be found at http://csqr.cass.cn/index.jsp (accessed on 12 October 2022), or from the corresponding author upon request.

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
