# Peer review of "The Influence of Subjective Socioeconomic Status on Life Satisfaction: The Chain Mediating Role of Social Equity and Social Trust"

_ijerph, 2022, doi:10.3390/ijerph192315652_

Round 1

Reviewer 1 Report

The authors of the paper set out to investigate the relationship between subjective socioeconomic status and life satisfaction, but also the mechanisms underlying social equity and social trust. They took into account the surveys of 17,217 Chinese citizens. The obtained results revealed a significant and positive relationship between the socioeconomic status of the subject and life satisfaction, as well as social equity and social trust.

The goals of the research were correctly formulated, the research hypotheses were properly defined, which ultimately led to interesting results.

However, there is a problem with the numbering of the headings:

– on page 1, the digit ‘1’ is missing in the heading. There should be: ‘1.1.’;

– on page 5, the number of the heading should be ‘4.’;

– on page 7, the number of the heading should be ‘5.’

Authors’ English is correct. The paper requires some editorial/technical amendments but not many linguistic corrections.

The literature presented in the references is relevant and up-to-date. There are some editorial errors or omissions and appropriate complements are required:

– In case of the source no. 13, the year should be 2022, not 2021. Besides, volume 61, issue 2, and pages 622-643 are missing.

– The source no. 13 seems quite dubious.

– In case of the source no. 17, the DOI number is missing. It should be: https://doi.org/10.1007/BF01079018.

– In case of the source no. 19, the DOI number is missing. It should be: 10.1037//0022-3514.59.6.1238. Besides, one name of the authors should be: ‘Buunk’ instead of ‘Bunnk’.

– In case of the source no. 20, the page range is missing. There should be: 815-823.

– In case of the source no. 28, the names have been mixed up with surnames. The start of the bibliographic record should go: ‘Yamamura, E., Tsutsui, Y., Yamane, C. et al.’

– In case of the source no. 29, the number of the article is missing. It should be: 1955.

– In case of the source no. 30, the DOI number is missing. It should be: https://doi.org/10.1111/j.2044-8309.1994.tb01008.x.

– In case of the source no. 33, the surname of one of the authors should be ‘Gil de Zúñiga’ instead of ‘Gil de Zuniga’.

– In case of the source no. 36, the number of the article is missing. It should be: 6410.

– In case of the source no. 47, the surname of one of the authors should be ‘Hayes’ instead of ‘Hays’.

– In case of the source no. 49, the DOI number is missing. It should be: https://doi.org/10.1177/001872675400700202.

– In case of the source no. 53, the issue number and the page range are missing. There should be: ‘23, 3485-3515’.

– Doubtful source no. 56. It cannot be found in any databases.

The results are presented in a clear way and the discussion conducted correctly, however the conclusions are quite lapidary.

Author Response

We thanks for your careful review. We did have some omissions in paragraph markers and references. Based on the issues you pointed out, we have made changes in the manuscript.

Secondly, we explain references no.13 and no.56, which we hope will answer the questions you raised.

For No. 13, we re-read the article based on your suggestion and found that we misinterpreted the author's meaning. Moreover, this article cannot be used as a theory for this paper. For this reason, we have re-reviewed the literature and re-explained why subjective socioeconomic status has a stronger association with life satisfaction.

No. 56 is from China National Knowledge Infrastructure. It is an online platform for readers at home and abroad to provide unified search, unified navigation, online reading and download services for all kinds of resources such as Chinese academic literature, foreign language literature, dissertations, newspapers, conferences, yearbooks, and tools.

Finally, we made certain language changes.

Reviewer 2 Report

The paper “apparently” seems complete and interesting despite the abstract needs to be completely redone and made more discursive. Reading it carefully, however, all the work is completely devoid of references to a concept that underlies both the SES and, above all, trust, that is social capital.

Social capital consists of the networks, norms, relationships, values and informal sanctions that shape the quantity and co-operative quality of a society’s social interactions. You must to know that three main types of social capital can be distinguished: bonding social capital (e.g. among family members or ethnic groups); bridging social capital (e.g. across ethnic groups); and linking social capital (e.g. between different social classes). In particular, social capital can be measured using a range of indicators but the most commonly used measure is trust in other people.

Thus, social capital may contribute to a range of beneficial economic and social outcomes including : high levels of and growth in GDP; more efficiently functioning labour markets; higher educational attainment; lower levels of crime; better health; and more effective institutions of government; different types of social capital are relevant to different economic and social outcomes e.g. bonding social capital is most important to health in early childhood and frail old age whereas bridging social capital is most

important in adult life when looking for employment; some of the empirical evidence on the importance of social capital for economic and social outcomes needs to be treated with caution because of the mis-specification or ambiguity of equations or models used to estimate its impact. But, overall, the evidence described in this paper from a range of soThe paper “apparently” seems complete and interesting despite the abstract needs to be completely redone and made more discursive. Reading it carefully, however, all the work is completely devoid of references to a concept that underlies both the SES and, above all, trust, that is social capital.

Social capital consists of the networks, norms, relationships, values and informal sanctions that shape the quantity and co-operative quality of a society’s social interactions. You must to know that three main types of social capital can be distinguished: bonding social capital (e.g. among family members or ethnic groups); bridging social capital (e.g. across ethnic groups); and linking social capital (e.g. between different social classes). In particular, social capital can be measured using a range of indicators but the most commonly used measure is trust in other people.

Thus, social capital may contribute to a range of beneficial economic and social outcomes including : high levels of and growth in GDP; more efficiently functioning labour markets; higher educational attainment; lower levels of crime; better health; and more effective institutions of government; different types of social capital are relevant to different economic and social outcomes e.g. bonding social capital is most important to health in early childhood and frail old age whereas bridging social capital is most

important in adult life when looking for employment; some of the empirical evidence on the importance of social capital for economic and social outcomes needs to be treated with caution because of the mis-specification or ambiguity of equations or models used to estimate its impact. But, overall, the evidence described in this paper from a range of sources using a variety of methods for the beneficial effects of social capital is impressive; social capital has potential downsides as well as potential benefits including: fostering behaviour that worsens rather than improves.

For all these reasons you should use this essential concept in your study if you want to make a real contribution to the scientific community. The paper in introduction, methods, results, discussion and conclusions, study design, etc. it is well structured, but without using the concept of social capital, the paper does not have much relevance.urces using a variety of methods for the beneficial effects of social capital is impressive; social capital has potential downsides as well as potential benefits including: fostering behaviour that worsens rather than improves.

For all these reasons you should use this essential concept in your study if you want to make a real contribution to the scientific community. The paper in introduction, methods, results, discussion and conclusions, study design, etc. it is well structured, but without using the concept of social capital, the paper does not have much relevance.

Author Response

We thank you for your careful review. Indeed social capital is a superordinate concept of socioeconomic status and trust, and trust is an important component of social capital. Therefore, we have added to the literature on social capital in the introduction and discussion section. In addition, we have reworked the abstract section based on your comments.

Round 2

Reviewer 2 Report

The paper with the corrections made is much more complete and interesting. For the future it is advisable to always keep in mind the importance and relevance that sociological concepts exercise in the different contexts of knowledge.